# Efficacy of Combined Use of Everolimus and Second-Generation Pan-EGRF Inhibitors in *KRAS* Mutant Non-Small Cell Lung Cancer Cell Lines

**DOI:** 10.3390/ijms23147774

**Published:** 2022-07-14

**Authors:** Renato José da Silva-Oliveira, Izabela Natalia Faria Gomes, Luciane Sussuchi da Silva, André van Helvoort Lengert, Ana Carolina Laus, Matias Eliseo Melendez, Carla Carolina Munari, Fernanda de Paula Cury, Giovanna Barbarini Longato, Rui Manuel Reis

**Affiliations:** 1Oncology Research Center, Barretos Cancer Hospital, Barretos 14784-400, Brazil; izabela.faria.tk@hotmail.com (I.N.F.G.); lsussuchi@gmail.com (L.S.d.S.); ahlengert@gmail.com (A.v.H.L.); anacarolinalaus@gmail.com (A.C.L.); melendezmatias@yahoo.com.ar (M.E.M.); carla.munari@elevescience.com.br (C.C.M.); fernandacury_7@hotmail.com (F.d.P.C.); giovanna.longato@usf.edu.br (G.B.L.); 2Life and Health Sciences Research Institute (ICVS) Medical School, University of Minho, 4710-057 Braga, Portugal; 3ICVS/3B’s-PT Government Associate Laboratory, 4710-057 Braga, Portugal

**Keywords:** pan-EGFR, *KRAS* mutations, mTOR, NSCLC, allitinib, afatinib, everolimus

## Abstract

Background: *EGFR* mutations are present in approximately 15–50% of non-small cell lung cancer (NSCLC), which are predictive of anti-EGFR therapies. At variance, NSCLC patients harboring *KRAS* mutations are resistant to those anti-EGFR approaches. Afatinib and allitinib are second-generation pan-EGFR drugs, yet no predictive biomarkers are known in the NSCLC context. In the present study, we evaluated the efficacy of pan-EGFR inhibitors in a panel of 15 lung cancer cell lines associated with the *KRAS* mutations phenotype. Methods: *KRAS* wild-type sensitive NCI-H292 cell line was further transfected with *KRAS* mutations (p.G12D and p.G12S). The pan-EGFR inhibitors’ activity and biologic effect of *KRAS* mutations were evaluated by cytotoxicity, MAPK phospho-protein array, colony formation, migration, invasion, and adhesion. In addition, in vivo chicken chorioallantoic membrane assay was performed in *KRAS* mutant cell lines. The gene expression profile was evaluated by NanoString. Lastly, everolimus and pan-EGFR combinations were performed to determine the combination index. Results: The GI_50_ score classified two cell lines treated with afatinib and seven treated with allitinib as high-sensitive phenotypes. All *KRAS* mutant cell lines demonstrated a resistant profile for both therapies (GI_50_ < 30%). The protein array of *KRAS* edited cells indicated a significant increase in AKT, CREB, HSP27, JNK, and, importantly, mTOR protein levels compared with *KRAS* wild-type cells. The colony formation, migration, invasion, adhesion, tumor perimeter, and mesenchymal phenotype were increased in the H292 *KRAS* mutated cells. Gene expression analysis showed 18 dysregulated genes associated with the focal adhesion-PI3K-Akt-mTOR-signaling correlated in *KRAS* mutant cell lines. Moreover, mTOR overexpression in *KRAS* mutant H292 cells was inhibited after everolimus exposure, and sensitivity to afatinib and allitinib was restored. Conclusions: Our results indicate that allitinib was more effective than afatinib in NSCLC cell lines. *KRAS* mutations increased aggressive behavior through upregulation of the focal adhesion-PI3K-Akt-mTOR-signaling in NSCLC cells. Significantly, everolimus restored sensibility and improved cytotoxicity of EGFR inhibitors in the *KRAS* mutant NSCLC cell lines.

## 1. Introduction

Lung cancer remains among the most common tumor worldwide, with the highest mortality, approaching 1.7–1.8 million deaths per year [1,2]. Only 20% of the cases are diagnosed in the early stages. The diagnosis is usually late and is made in the advanced stages, leading to inefficient treatment [3]. Smoking is the major risk factor, and it is estimated that 83% of new lung cancer cases will be associated with this habit [4]. Other risk factors include occupational exposure, long-term exposure to polluted air, chronic obstructive lung disease, and family history [5].

According to the histological context, lung cancer can be mainly classified into small cell lung cancer (SCLC) and non-small cell lung cancer (NSCLC), with incidences of 15% and 85%, respectively. In addition, NSCLC receives histologically two subclasses: adenocarcinomas, comprising about 40% of the NSCLC cases, and squamous cell carcinomas. Complete surgical resection is amenable in 70% of adenocarcinoma [6], 25 to 30% in squamous cell carcinomas, and just over 10–15% in large cell carcinomas or undifferentiated tumors [7].

Genetic alterations are associated with NSCLC tumorigenesis, and currently guide the therapeutic strategies [8]. *EGFR*, *KRAS* gene mutation, and *ALK* translocation are some of the most common mechanisms of carcinogenesis in NSCLC [9]. EGFR alterations are detected in up to 40% of NSCLC patients. Therefore, gefitinib and erlotinib are the first-generation of reversible EGFR-TKIs, for *EGFR* mutated (L858R and or exon del 19) patients [10]. The second-generation EGFR-TKIs are irreversible drugs such as afatinib, approved for patients treated with the erlotinib and gefitinib with drug-resistant mutations T790M, which is found in approximately 60% of patients with acquired resistance [11]. Less explored allitinib, also known as AST1306, is a potent anti-EGFR with irreversible action on EGFR family members (EGFR, HER2, and HER4), including *EGFR* mutant T790M/L858R and exhibits antitumor activity in pre-clinical and clinical trials [12]. Cytotoxicity analysis of allitinib has been conducted on an extensive panel of tumor cell lines and displayed cytotoxicity effect in head and neck, esophageal, melanoma, and lung cancer-derived cell lines [13].

The emergence of a third generation of EGFR-TKI is based on the efficacy and safety study of osimertinib, which showed excellent performance in second-line treatment and offered more survival benefits for patients with the T790M mutation [14]. Osimertinib clinical trials (NCT02296125) have shown superior results of osimertinib compared with standard EGFR-TKIs in the first-line treatment of *EGFR* mutation-positive advanced NSCLC [14]. In addition, results from the phase 3 AURA3 trial demonstrated the superiority of osimertinib over standard platinum-based treatment of NSCLC patients with *EGFR* T790M mutation relapsed or refractory to first-line EGFR TKI therapy, thus definitively establishing this third-generation TKI as current standard treatment [15]. Other third-generation EGFR TKIs, such as EGF816, olmutinib, PF-06747775, YH5448, avitinib, and rociletinib, continue to be investigated in clinical trials phases to prove their therapeutic efficiency alone or in combination [16].

*KRAS* mutations are mostly located in codons 12 and 13 and can occur between 10 to 25% in European patients, approximately 10% in East Asians [17], and in 25% of the admixture Brazilian NSCLC population [18]. The oncoprotein KRAS was for decades considered an “undruggable” target, and recently, sotorasib (AMG510) was approved by the United States Food and Drug Administration (FDA) for the treatment of adult patients with specific *KRAS* G12C-mutated locally advanced or metastatic NSCLC [19]. Moreover, adagrasib (MRTX849), also targeting *KRAS* G12C-mutated tumors, is currently being evaluated in phase 1/2 with results resembling the efficacy of sotorasib [20]. A recent update by the KRYSTAL-1 trial showed tolerable levels (600 mg twice daily). It demonstrated durable (16.4 months) clinical activity in NSCLC patients *KRAS* G12C mutant [21], and a current phase 3 trial (NCT03785249) evaluating adagrasib as monotherapy versus docetaxel in *KRAS* mutant NSCLC patients is ongoing [22].

Despite these developments, NSCLC patients’ platinum-refractory with non-*KRAS* G12C mutation is still a therapeutic challenge, and the high cost of these anti-KRAS inhibitors constitutes an important barrier in low and middle-income countries. Interestingly, it has been shown that *KRAS* mutant patients can exhibit high EGFR expression levels [23]. Therefore, our study explored the role of EGFR inhibitors (afatinib and allitinib) and conducted a comprehensive in vitro and in silico analysis of the molecular mechanisms triggered by *KRAS* mutations in a panel of lung cancer cell lines.

## 2. Results

### 2.1. Comparative Efficacy of Afatinib and Allitinib in Lung Cancer Cell Lines

We used a large panel of lung cancer cell lines to compare the efficiency of second-generation irreversible EGFR inhibitors afatinib and allitinib to determine the half-maximal inhibitory concentrations (IC_50_) values. Of the cell lines evaluated, three harbor EGFR mutations (NCI-H1975, NCI-H827, and PC9) and four *KRAS* mutations (SK-LU-1, A549, NCI-H358, and NCI-H727) (Table 1). The afatinib cytotoxic effect revealed low IC_50_ values ranging from 0.34 ± 0.03 to 0.72 ± 0.05 µM, for EGFR mutated cells, while allitinib IC_50_ values were even smaller, from 0.21 ± 0.09 to 0.31 ± 0.07 µM (Table 1). Notably, afatinib and allitinib showed smaller IC_50_ average values in *KRAS* mutant cell lines (2.9 ± 2.6 and 2.1 ± 1.3 µM) compared to erlotinib and lapatinib inhibitors (30.4 ± 20.3 and 22.8 ± 23.6). Allitinib was the most potent EGFR inhibitor in the cell line panel, with an IC_50_ value of 1.8 ± 1.9 µM, while afatinib showed values of 2.4 ± 2.3 µM (Table 1).

According to GI classification, seven cell lines showed a sensitive phenotype (HS: high sensitivity + MS: moderate sensitivity) to afatinib, and eight presented a resistant phenotype. Meanwhile, ten cell lines were classified as sensitive to allitinib and five cell lines were classified as resistant (Figure 1A,B and Appendix A). Alternatively, we tested the cytotoxic potential of both inhibitors with the ApoToxiGlo assay, which analyzes cell membrane integrity by fluorescence intensity. These results revealed that allitinib has high cytotoxicity for mutant EGFR cell lines (PC9 and NCI-H1975), low effect for *KRAS* mutant cell line (A549), and shows cytotoxic activity to H292 KRAS wild-type cell line (Appendix A). To evaluate cellular proliferation curves, we exposed *KRAS* mutant cell lines (A549), wild-type (H292), and EGFR mutant cells (H1975 and PC9) to a fixed concentration of both inhibitors (0.2 µM) for 5 days. As shown in Appendix A, none of the inhibitors decreased proliferation in the *KRAS* mutant cell line A549. However, a significant proliferation curves inhibition in EGFR mutant cell lines (H1975 and PC9) and EGFR wild-type (H292) was observed. We also investigated the intracellular pathway changes by Western blot to determine the biological effect of irreversible EGFR inhibitors (Figure 1C). Cell lines were exposed to inhibitors (0.2 µM) for two hours and then stimulated to EGF ligand (10 ng/mL) for 10 min. We found a reduction in EGFR phosphorylation levels in PC9, H1975 (EGFR mutant), and H292 (EGFR wild-type) cell lines. In addition, we observed a significant decrease in ERK and AKT phosphorylation only in EGFR mutant PC9 and H1975 cell lines. Notably, the A549 cell line showed a minimum decrease in EGFR, AKT, and ERK phosphorylation levels. PC9, H1975, and H292 cell lines were sensitive to both EGFR inhibitors and showed a severe reduction in phosphorylated AKT proliferation markers following allitinib exposure (Figure 1C). Finally, PARP activation and subsequent cleavage were detected with greater intensity in H292 (EGFR wild-type), PC9 and H1975 (EGFR mutant), and A431 (sensitive control), although low cleaved-PARP levels were detected in *KRAS* mutant cell lines (A549 and SKLU-1) (Figure 1C).

### 2.2. Phenotype Modulation and Resistance to EGFR Inhibitors Caused by KRAS Mutations

To determine the biological impact of *KRAS* mutations in NSCLC, gene-edited lung cancer cell lines H292-WT, H292-G12D, and H292-G12S were used. Primarily, adhesion ligand proteins were analyzed in both mutant cells, and we found a significant increase in cell–protein adhesion in KRAS mutant cells compared to wild-type H292 cells (Figure 2A). In addition, we detected an adhesion inhibition after allitinib exposure (Figure 2A,B). A colony formation assay was conducted to evaluate cellular clonogenicity, and H292 wild-type cells exhibited fewer viable colonies than mutant cells, and showed their sensibility to both anti-EGFR agents (Figure 2C,D). *KRAS* mutant cell lines showed a significant increase in the number of colonies formed after both treatments. In addition, we conducted an invasion assay to access the cell motility and invasiveness capacity toward a chemo-attractant gradient. Each cell line was exposed to the EGFR inhibitors, and we found an increased invasion of *KRAS* mutation cells lines (Figure 2E,F) compared to *KRAS* wild-type alone or treated with EGFR inhibitors.

### 2.3. Molecular and Genetic Associated with KRAS Mutation NSCLC Cell Lines

To characterize the molecular profile associated with *KRAS* mutant status, we initially compared the MAPKs profile between H292-WT and *KRAS* mutated H292-G12D and H292-G12S cells using a Human Phospho-Mitogen-activated Protein Kinase (MAPK) assay. Basal protein expression levels were altered, due to *KRAS* mutation insertion, and the main phospho-protein targets altered were AKT isoforms, ERK1 and 2, JNK2, HSP27, panJNK, P38δ, RSK1, RSK2, and mTOR (Figure 3A,B). We confirmed that proliferation, Wnt/β-catenin, AKT isoforms, and epithelial–mesenchymal transition markers were altered in *KRAS* mutation presence and remained conserved in specific Western blot for both cell lines. Additionally, we observed an increase in phospho-mTOR expression in *KRAS* mutant cell lines that was validated by Western blot analyses (Figure 3C). Furthermore, the Wnt/β-catenin pathway, which regulates morphogenesis, adhesion, and migration, was evaluated by Western blot. The *KRAS* mutant cell lines showed decreased C-MYC, C-JUN, CD44, MET, and TCF1/TCF7 protein expression (Figure 3D). Moreover, EMT protein analysis showed an increase in the mesenchymal vimentin, SLUG, and N-cadherin markers, whereas SNAIL and E-cadherin proteins involved in cell–cell contacts, were decreased (Figure 3E). The CAM assay was performed to evaluate tumor growth of H292 *KRAS* mutant cell lines, and we found a significant increase in tumor perimeter of both *KRAS* mutant cells, compared to wild-type (Figure 3F and Appendix A). Next, we investigated mesenchymal markers expression in both cells, which showed a positive mRNA expression and protein levels for SLUG, VIM, N-cadherin, and interestingly, *KRAS* mutant cell lines, increased the metalloproteinases expression of MMP-9, MMP-24, and MMP-2 (Appendix A).

Additionally, using the NanoString platform, we evaluated the gene expression pattern of 13 critical cancer-related pathways. Significant alterations of genes involved in the focal adhesion-PI3K-Akt-mTOR-signaling pathway in H292 *KRAS*-G12D (43 genes) and H292 *KRAS*-G12S (32 genes) compared to *KRAS* wild-type cells (Figure 4A,B) was observed. Among these genes, 18 were differentially expressed in both H292 *KRAS*-G12D and H292 *KRAS*-G12S (Appendix A). Of note, the upregulation of SPP1, EFNA2, and the downregulation of THBS1, ITGA4, TNC, and FGF21, exhibited more remarkable fold changes in both mutant cell lines. In silico interaction network analysis using these 18 key genes altered in the *KRAS* mutant cells from the STRING database (Version: 11.0, https://string-db.org accessed on 9 February 2021) revealed a tight network for the mTOR-mediated signaling pathway with a PPI (protein-protein interaction) enrichment of *p*-value: <1.0 × 10^−16^ (Figure 4C).

Due to the increased protein expression of the mTOR signaling pathway, we conducted an additional prognostic analysis (data set GSE13213), and we found a significant correlation between mTOR (mRNA) overexpression with low overall survival rates in patients with lung adenocarcinoma (HR = 1.16, 95%, *p* = 0.04). Kaplan–Meier curve (Figure 4D) revealed that mTOR high mRNA levels correlate with poorer prognostic and survival, suggesting proactive involvement of this gene in the malignancy in lung adenocarcinoma.

### 2.4. mTOR Inhibition Restores Sensitivity to Pan-EGFR Inhibitors in KRAS Mutation NSCLC Cell Lines

Since the mTOR pathway was a major pathway upregulated by *KRAS*-induced mutations, and these cells also exhibited high EGFR expression, we next performed a therapeutic combination with everolimus (anti-mTOR) and afatinib or allitinib (pan-EGFR inhibitors) drugs (Figure 5A–C). H292 *KRAS* wild-type did not show a decrease in the IC_50_ values after everolimus combination, whereas *KRAS* mutant cells reduced the IC_50_ values from 5.8 ± 0.7 to 3.9 ± 0.5 μM for H292 *KRAS*-G12D, and from 8.5 ± 0.5 to 2.9 ± 0.7 μM for H292 *KRAS*-G12S after afatinib and everolimus combination (Appendix A). Similarly, IC_50_ values were decreased from 2.9 ± 1.2 to 1.0 ± 0.4 μM for H292 *KRAS*-G12D, from 4.2 ± 1.1 to 0.9 ± 0.7 μM for H292 *KRAS*-G12S after allitinib and everolimus combination (Appendix A). We further evaluated the potential combinatorial values to determine the IC_50_ for each EGFR inhibitor alone, or in combination with everolimus (1 μM fixed dose). As depicted in Figure 5, we found mutually synergically drug association (CI < 1), except for afatinib plus everolimus combination in H292 wild-type cell. No cytotoxic effect was observed in H292 wild-type and *KRAS* mutant cell lines after exposure to everolimus alone (Appendix A).

## 3. Discussion

In the present study, we compared the in vitro efficacy of EGFR inhibitors (afatinib and allitinib) and conducted a comprehensive analysis of the molecular mechanisms triggered by *KRAS* mutations in a panel of lung cancer cell lines.

It was observed that allitinib showed a superior cytotoxicity effect in a panel of NSCLC cell lines, including those harboring secondary *EGFR* mutations (T790M). Additionally, allitinib showed greater GI levels than afatinib, and superior proliferation inhibition in EGFR mutant cells. A large-scale study conducted by our group previously reported similar cytotoxic effects to other tumor types, such as melanoma, pancreas esophagus cell lines, where the majority (70%) of the *KRAS* mutant cell lines were classified as allitinib resistant [13]. These findings corroborate the resistance role of *KRAS* mutations in clinical trials (NCT04671303) [24].

In addition, we showed complete inhibition of the phospho-ERK and phospho-AKT activity only in *EGFR* and *KRAS* wild-type cell lines in the presence of both pan-EGFR inhibitors. On the contrary, these pathways are not inhibited in the *KRAS* mutant cells. It is well known that the ERK-MAPK pathway plays a central role in oncogenic KRAS-driven malignant phenotypes of NSCLC [25]. Therefore, some studies demonstrated an inverse association suggesting that *KRAS* oncogenic desensitizes cells to EGF at the level of the EGFR. In contrast, a recent study reported that cells (A549 gene-editing) with concomitant *KRAS*-*EGFR* mutation exhibited higher sensitivity to EGFR-tyrosine kinase inhibitors (TKIs). Indeed, EGFR phosphorylation is strongly compromised in the presence of oncogenic *KRAS* [26], as observed in the A549 and SKLU-1 cells. EGFR inhibitors were more effective for EGFR mutant cell lines (PC9 and H1975) and wild-type (H292) revealed by PARP increased level compared to *KRAS* mutant cells.

In our study, *KRAS* mutation promoted an aggressive phenotype in the H292 gene-edited cell line, as well as an increase in cell–protein adhesion, superior clonogenicity profile, and higher invasiveness rates, and stronger metastatic abilities. KRAS mutant cells showed an increased cell-to-extracellular matrix interaction and concomitant integrin subunit beta 8 gene (ITGB8) overexpression, which plays an important role in cell adhesion to protein matrix components and has been associated as a diagnostic biomarker for lung NSCLC patients [27]. Furthermore, high expression levels of ITGB8 have already been associated with angiogenesis in other tumor types, such as glioblastoma [28]. Our findings open new opportunities to explore other receptors such as FAK (focal adhesion kinases) that have already been related in *KRAS* mutant patients. FAK may be surrogate markers of aberrant *KRAS* signaling found in aggressive phenotype in lung cancer [29]. A similar effect was observed in the *KRAS*-mutated adenocarcinoma cell line model (A549-FL), used to examine the metastasis mechanisms that identified epithelial–mesenchymal transition (EMT) markers [30]. We found that EMT markers were affected at gene and protein expression levels, in *KRAS* mutant cell lines, with downregulation of E-cadherin, and upregulation of N-cadherin, vimentin, and SLUG. Importantly, SLUG overexpression has been identified as the key factor responsible for resistance to MEK1/2 inhibition and increased metastasis in other cancer types [31]. Additionally, SLUG combines with the loss of E-cadherin expression, which frequently occurs during tumor metastasis due to regulation of the cell adhesive activity [32] mainly known to induce epithelial-to-mesenchymal transition.

The crosstalk of oncogenic pathways is present in NSCLC tumorigenesis. A recent study found that the activation of Wnt/β-catenin alone does not promote tumor development but concomitant activation of Wnt/β-catenin signaling and *KRAS* mutation (G12D) led to an increase in tumor number and size [33]. A similar increase in Wnt/β-catenin signaling was reported in our study, where H292 *KRAS* mutant cells showed an increase in TCF1/TCF7 transcription factors, which regulate important genes such as c-*myc*, that is responsible for the proliferation stimulus and apoptosis decision and matrix metalloproteinase-7 expression [34]. Based on these findings, we hypothesize that the increase in tumor perimeter found in CAM analyses, may be related to EMT markers caused by the *KRAS* mutant gene in H292 edited cells.

Molecular analyses showed a critical difference between wild-type and *KRAS* mutant cells, as a differentiated pattern of MAPK pathway proteins. Both *KRAS* mutation cells display strong phosphorylation of several proteins, such as AKT, ERK1/2, JNK, HSP27, and predominantly mTOR protein, compared to wild-type cells. The PI3K/AKT/mTOR axis is involved in tumor survival, proliferation, and distant metastasis, supporting the development of target therapies [35]. A recent study demonstrates a positive association between elevated expression of phospho-AKT, phospho-mTOR with metastasis, and poor prognosis of NSCLC patients [36]. Another consequence of PI3K/AKT/mTOR activations is focal adhesion kinases (FAK) dysfunction. FAK is a non-receptor protein tyrosine kinase identified as a critical signaling molecule mediating host–tumor crosstalk, affecting cell adhesion, invasion, angiogenesis, metastasis, and is frequently overexpressed in various malignancies, including lung cancer [37,38].

In the comparative analysis of the gene expression profile, we found 18 genes simultaneously differentially expressed in both *KRAS* mutant cell lines (G12D and G12S). Superior fold change was detected in secreted phosphoprotein 1 (*SPP1*) in *KRAS* mutant cell lines. The *SPP1* overexpression is associated with aggressive phenotypes of lung cancer [39] and recently was correlated with afatinib resistance and shorter overall survival in NSCLC patients [40]. Moreover, in a mouse model of tobacco carcinogen-induced, the overexpression of *SPP1* was deleterious for mice harboring *KRAS* G12D during the early stages of carcinogenesis [41]. In addition, the *EFNA2* gene, also overexpressed in both *KRAS* mutant cell lines, is considered a ligand of Ephrin family receptors, and it is involved in the WNT/ß-catenin signaling pathway by TCF/LEF 1 transcriptional complex activation [42], which were also increased in our *KRAS* mutant cells.

*KRAS* mutant cells presented a significantly *ITGB4* and *ITGA6* downregulation. These genes are members of the integrin family of the adhesion receptors that play a major role in epithelial organization and function and display a very high adhesive strength for laminin matrices regulating metastasis development [43]. Based on this biological function, we hypothesize that the abrupt downregulation of *ITGB4/ITGA6* genes may be related to the aggressiveness, increased invasion, and changes in adhesion found in mutated *KRAS* cells. We also found a significant loss in *IL2RA* and *TNC* expression in *KRAS* mutant cells, and little is known about the role of these two genes in lung carcinogenesis.

Importantly, we observed that the mTOR pathway was significantly enriched in *KRAS* mutant cells, and the in-silico analyses indicated a significant association between *mTOR* overexpression and worse overall survival in NSCLC patients. Likewise, a recent study confirmed that phosphorylated-mTOR protein is associated with metastasis and poor prognosis of NSCLC patients after surgical resection [36]. Therapeutically, mTOR inhibitor monotherapy has met limited clinical success [44]. Moreover, in the past, in vitro and clinical evidence showed toxicity between co-target agents everolimus and gefitinib (NCT00456833) [45]. However, we found that the combination of mTOR inhibitor (everolimus) with afatinib and allitinib, a new generation of EGFR inhibitors that require lower doses to become effective, therefore with putative less toxicity, shown a synergistic association mainly in the *KRAS* mutant cells. Similar dual inhibition encouraging results are reported in other cancer types, such as triple-negative breast cancers [46]. Nevertheless, future in vivo studies and clinical trials need to test this putative novel therapeutic approach for NSCLC patients harboring *KRAS* mutations.

## 4. Materials and Methods

### 4.1. Cell Lines and Cell Culture Conditions

Fifteen lung cancer cell lines from different anatomic sites were used in this study, namely NCI-1975; COR-L23; COR-L105; LUDLU-1; NCI-H322; NCI-H358; NCI-H727, NCI-H2228 NCI-H827; PC9; SK-MES-1; SK-LU-1; A549, H292, and Calu-3 (Appendix A). We also used genetically modified human cell lines H292-WT (wild-type), and *KRAS* mutant H292-G12D and H292-G12S previously developed by our group [13]. The epidermoid carcinoma cell line A431 was used as sensitive control due to high levels of EGFR expression. Cell lines were maintained in Dulbecco’s modified Eagle’s medium (DMEM) or RPMI 1640, containing 10% fetal bovine serum (FBS), 2mM glutamine, and 1% penicillin/streptomycin. Cells were incubated in a humidified atmosphere of 5% CO_2_ at 37 °C. Cell culture reagents were purchased from Sigma-Aldrich (St. Louis, MO, USA), and to avoid the misidentified and/or cross-contamination, cell lines were authenticated by STR analysis [13]. Culture supernatants were regularly tested for mycoplasma contamination by MycoAlert^TM^ PLUS Mycoplasma Detection Kit (Lonza, Walkersville, MD, USA).

### 4.2. Pharmacological Agents

Afatinib (Cat. No. S1011), allitinib (Cat. No. S2185), lapatinib (Cat. No. S2111), erlotinib (Cat. No. S1023), and everolimus (Cat. No. S1078) were purchased from Selleck Chemicals (Houston, TX, USA). All drugs were diluted in DMSO at 10 mM and stored at −20 °C for future use. Everolimus was dissolved in distilled water at 10 mM. In all experiments, DMSO or water were used as control vehicles at a final concentration of 1% (*v*/*v*) in all experiments.

### 4.3. Cytotoxicity, Growth Inhibition, and Proliferation Analyses

According to the manufacturer’s instructions, cell viability was determined 72 h after drug treatments, using the colorimetric CellTiter 96^®^ AQueous One Solution Cell Proliferation Assay (Promega, Madison, WI, USA) as previously reported [13]. To assess the cytotoxicity, a total of 5 × 10^3^ cells were plated in 96-well plates in DMEM or RPMI-10%, Fetal bovine serum (FBS) allowed to adherence overnight in a CO_2_ atmosphere. Then, cells were treated with increased concentrations of afatinib, allitinib, and everolimus (0.2; 0.4; 0.6; 0.8; 1.0; 2.0; 5.0 μM) in DMEM 0.5% FBS. Absorbance was measured at 490 nm using the Varioskan Flash multimode reader (Thermo Scientific, Oy, Vantaa, Finland). Results were normalized with DMSO control values. GraphPad Prism software (Version 9.0, San Diego, CA, USA) was used to calculate calculated IC_50_ values, using a nonlinear regression analysis. The cytotoxicity was measured by ApoToxiGLo kit (Promega, Madison, WI, USA), according to the manufacturer’s instructions. In this assay, cell lines were plated in 96-well plates in DMEM or RPMI 10% allowed adherence overnight in a CO_2_ atmosphere and treated with fixed concentrations (0.5, 5, and 10 µM) for 72 h. Mean growth inhibition (GI) values were calculated at a fixed concentration of 1 µM (allitinib and afatinib). The A431 cell line was used to determine the cutoff sensibility value due to high EGFR expression levels and our previous allitinib sensitivity results [13]. Cell lines were classified as highly sensitive (HS) if GI > 60%, moderately sensitive (MS) if GI 40–60% and resistant if GI < 40%, as previously described [13,47]. All experiments were performed in triplicates at least three times. Drug interactions were analyzed using CalcuSin software (Calcusin, Version 2.0, Biosoft, Cambridge, UK) to determine combination index (CI) and the effect was classified in synergy (CI < 1.0), antagonism (CI > 1.0), and additivity (CI = 1.0) [48].

### 4.4. Cellular Adhesion and Invasion Assay

To assess the cellular adhesion attachment event, we used the CAM assay as previously described [49]. The 96-well plate was coated for 24 h with a solution containing PBS, BSA (bovine serum albumin, 10 μg/mL, Sigma-Aldrich), Matrigel^®^ (1:10 in PBS). After 24 h, the excess liquid was removed, and cell plates were incubated with 100 μL/well of 0.1% BSA for 2 h and washed with PBS. Wild-type and *KRAS* mutant cell lines were seeded in duplicate and incubated at 37 °C in a 5% CO_2_ humidified atmosphere for 2 h. Non-adherent cells were rinsed with PBS solution. Adhered cells were fixed with 10% of trichloroacetic acid (TCA) and at last, stained with crystal violet solution (0.5%). Additionally, crystal violet dye was dissolved in 10% of acetic acid solution and absorbance was obtained at 590 nm using a Varioskan Flash multimode reader (Thermo Scientific, Finland). Absorbance values of the samples were plotted in GraphPad Prism software to create relative colony growth. All experiments were performed in triplicates at least three times.

### 4.5. Clonogenic Assay

Wild-type and *KRAS* mutant cell lines (H292-WT; H292-G12D and H292-G12S) were seeded in duplicate into 24-well plates (0.7 × 10^3^ cells per well) and allowed to adhere overnight in media with 10% FBS. Subsequently, cells were treated with fixed concentrations of afatinib and allitinib (0.2 µM) for 21 days. After, cell lines were fixed with methanol for 10 min. The colonies were stained with crystal violet solution (0.5%) and the colony images were obtained using an optical system, Olympus SZX7. Moreover, crystal violet dye was dissolved in 10% of acetic acid solution and absorbance was obtained at 590 nm using Varioskan Flash multimode reader (Thermo Scientific, Finland). Absorbance values of the samples were plotted in GraphPad Prism software to create relative colony growth. All experiments were performed in triplicates at least three times.

### 4.6. Western Blot Analysis and Human MAPK Arrays

EGFR inhibition and intracellular pathways signaling were evaluated in *EGFR* mutant cell lines (PC9 and H1975), *KRAS* wild-type cell line (H292), and *KRAS* mutant cell lines (A549 and SKLU-1). Cell lines were plated in 6-well plates (5.0 × 10^5^) in DMEM or RPMI; 10% FBS allows for adherence overnight in a CO_2_ atmosphere. After 24 h, cell lines were rinsed in ice-cold PBS, then scraped and lysed in lysis buffer (50 mM Tris pH 7.6–8, 150 mM NaCl, 5 mM EDTA, 1 mM Na3VO4, 10 mM NaF, 10 mM sodium pyrophosphate, 1% NP-40, and protease cocktail inhibitors). In total, 30 μg of total protein was resolved by 10% SDS-PAGE and immediately transferred to nitrocellulose membranes in TransBlot Turbo transfer (Bio-Rad, Bio-Rad, Hercules, CA, USA). Primary antibody was purchased from Cell Signaling Technology (CST, Danvers, MA, USA): human total EGF Receptor (Cat. No. D38B1; RRID:AB_10692501), pEGFR-Tyr1068 (Cat. No. D7A5; RRID:AB_10828604), p44/42 MAPK (Cat. No. 137F5; RRID:AB_10693607); p.p44/42 MAPK-Thr202/Tyr204 (Cat. No. D13.14.4E; RRID:AB_10694057); AKT(pan) (Cat. No. C67E7; RRID:AB_915783); pAKT-Ser473 (Cat. No. D9E; RRID: AB_2224726); total PARP (Cat. No. 9532; RRID: AB_659884); AKT1(Cat. No. 2938; RRID: AB_915788); AKT2 (Cat. No. 3063; RRID:AB_2225186); AKT3 (Cat. No. 8018; RRID: AB_10859371); phospho-mTOR (Cat. No. 5536; RRID: AB_10691552); c-MYC (Cat. No. 5605; RRID: AB_1903938); c-JUN (9165; RRID: AB_2130165); CD44 (Cat. No. 3570; RRID: AB_2076465); Ciclin D1 (Cat. No. 2978; RRID: AB_2259616); Met (Cat. No. 8198; RRID: AB_10858224); TCF1/TCF7 (Cat. No. 2203; RRID: AB_2199302); Snail (Cat. No. 3879, RRID: AB_2255011); Slug (Cat. No. 9585; RRID:AB_2239535); E-caderin (Cat. No. 3195; RRID: AB_2291471); N-Caderin (Cat. No. 13116; RRID: AB_2687616); α-Smoth (Cat. No. 19245 RRID: AB_2734735); Vimentin (Cat. No 5741; RRID: AB_10695459) and β-tubulin (Cat. No. 2128 RRID: AB_823664) as endogenous control. Primary antibodies were diluted in TBS-T solution at 1:1000 and incubated overnight at 5 °C. Then, membranes were incubated with anti-rabbit secondary antibody Antirabbit (Cat. No. 7074; RRID: AB_2099233) at dilution 1:5000. Immune detection was performed using Pierce™ ECL Western (Thermo Scientific, Rockford, IL, USA), in automatic ImageQuant mini LAS4000 chemiluminescence systems (GE Healthcare, Fairfield, CT, USA). Western blot experiments were performed three times. A total of the 750 μg of fresh protein lysates was used to perform Human Phospho-Mitogen-activated Protein Kinase (MAPK) Antibody Array (Cat. No. ARY002B; R&D Systems, Minneapolis, MN, USA). The protein content of each cell lines was incubated overnight at 4 °C with nitrocellulose membranes each containing 24 different kinases antibodies spotted in duplicate on a nitrocellulose membrane. Then, phospho-MAPKs membranes were incubated with a pan anti–MAPKs-HRP antibody for 2 h at room temperature. Blot detection was carried out by chemiluminescence reagent in ImageQuant LAS 4000 mini (GE Healthcare). Densitometric analyses of dot or bands were measured using ImageJ software (version 9.0, NIH, Bethesda, MD, USA) [50].

### 4.7. Chick Chorioallantoic Membrane (CAM) Assay

To assess in vivo *KRAS* mutant cell lines tumor proliferation, we used the CAM in vivo assay as previously described [50]. Fertilized chicken eggs were incubated at 37 °C and 70–80% relative humidity, and on day 3 of development, a window was cut through the shell of each egg, returning to the incubator. A total of 2 × 10^6^ cells were suspended and immersed in Matrigel solution, then incubated inside CAM on day 9 of development. After 4 days, the 3D tumor model was observed, and daily growth was monitored throughout the development. At the end of the assay, the tumor was photographed in ovo (14 and 17 days of development) and ex ovo (17 day of development) using a stereomicroscope (Olympus SZX7), and the perimeter of the tumors was measured using ImageJ software (version 9.0). The results were expressed as the mean perimeter ± SD.

### 4.8. Real-Time Quantitative PCR

Real-time quantitative PCR (RT-qPCR) was performed for specific genes linked to epithelial–mesenchymal transition, migration, and invasion using the GoTaq^®^ DNA Polymerase system (Promega) according to the manufacturer’s instructions. Real-time PCR was performed using a Step One Plus instrument (Life Technologies, Carlsbad, CA, USA). PCR conditions were 95 °C for 2 min to activate DNA polymerase, followed by 40 cycles at 94 °C for 15 s and specific annealing temperature of each primer (Appendix A). The difference in cycle threshold value (Ct) of H292 *KRAS* wild-type versus internal control (ΔCt) was used to determine gene expression in the *KRAS* mutated cells.

### 4.9. mRNA NanoString^TM^ Analysis

Gene expression analysis on H292-WT, H292-*KRAS*-G12D, and H292-*KRAS*-G12S were performed using the NanoString nCounter PanCancer Pathways panel (770 gene transcripts) according to the manufacturer’s instruction and as described (manuscript accepted in Scientific Reports) (NanoString Technologies, Seattle, WA, USA). This panel assesses 13 canonical pathways (Notch, Wnt, Hedgehog, chromatin modification, transcriptional regulation, DNA damage control, TGF-beta, MAPK, STAT, PI3K, RAS, cell cycle, and apoptosis). Briefly, 100 ng aliquots of RNA quantified by Qubit 2.0 System (Life Technologies, Carlsbad, CA, USA) were hybridized with capture and reporter probes and hybridization buffer in a total volume of 15 μL, incubated at 65 °C for 20 h. After codeset hybridization, the samples were purified, and RNA-probe complexes were immobilized to a cartridge using the NanoString nCounter Prep Station (NanoString Technologies, Seattle, WA, USA) for 4 h. Finally, the cartridges containing immobilized and aligned RNA-probes complexes were scanned in the nCounter Digital Analyzer (NanoString Technologies) using 280 FOVs (Fields of View), and image data were subsequently generated using the high-resolution setting. Raw data quality control assessments were performed with nSolver Analysis Software version 4.0 with the default settings (NanoString Technologies, Seattle, WA, USA). Quantile normalization and differential expression were performed within NanoStringNorm package (v1.2.1.1. NanoString Technologies, Seattle, WA, USA) [51] in the R statistical environment (v3.6.3. RStudio Inc, Boston, MA, USA). The normalized log2 mRNA expression values were used for subsequent data analysis. Differential expression, hierarchical clustering, and pathway analysis were performed in the Rosalind^®^ platform, available at https://rosalind.onramp.bio accessed on 15 October 2021. Genes with fold change (FC) ≥ ±2 and *p* < 0.05 were considered significant.

### 4.10. PrognoScan Database Analysis

For meta-analysis of the prognostic value of the *mTOR* gene in adenocarcinomas, the PrognoScan database was used. This database is an extensive collection of publicly available cancer microarray datasets with clinical annotation and the biological relationship between gene expression and prognosis (http://www.prognoscan.org/ accessed on 15 October 2021) [52]. This database is a robust platform for evaluating potential tumor markers and therapeutic targets. The association between key genes was analyzed using PrognoScan, and a *p*-value < 0.05 was considered statistically significant.

### 4.11. Statistical Analysis

Single comparisons between the different conditions studied were made using Student’s *t*-test, and differences between groups were tested using a two-way analysis of variance. Statistical analysis was carried out using GraphPad Prism version 9. The level of significance in all the statistical analyses was set at *p* < 0.05.

## 5. Conclusions

In summary, we showed a higher cytotoxicity effect of allitinib in a representative panel of lung cancer, and the sensibility was associated with *KRAS* mutant status. We elucidate the phenotypic and molecular mechanisms activated by *KRAS* mutations and identified novel genes, such as *SPP1*, *EFNA2*, *ITGA6*, *ITGB8, ITGB4*, *IL2RB*, and *TNC*, associated with PI3K/AKT/mTOR pathway activation. Our findings provide preclinical evidence supporting the rationale for combining the mTOR inhibitor with afatinib or allitinib for NSCLC treatment in *KRAS* mutant patients.

## Figures and Tables

**Figure 1 ijms-23-07774-f001:**
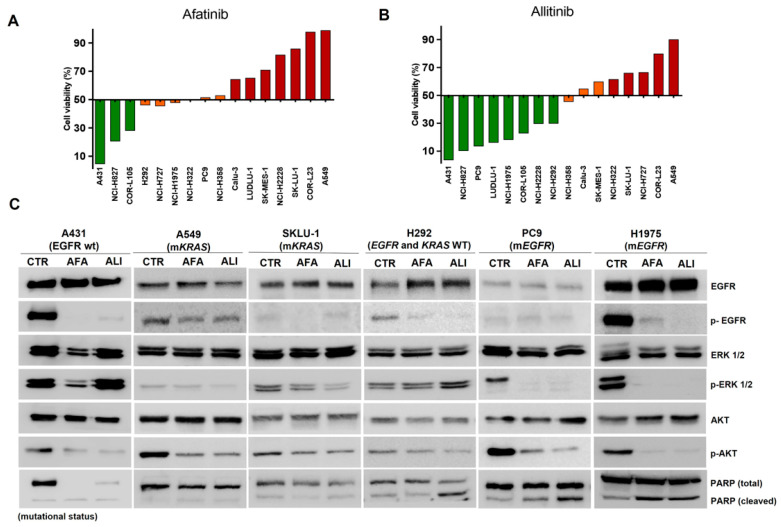
The comparative effect between afatinib and allitinib. The growth inhibition score (GI) of NSCLC cells was calculated for afatinib (**A**) and allitinib (**B**) at 1000 nM, classified as highly sensitive-HS (green bars), moderate sensitivity-MS (orange bars), and resistant-R (red bars). (**C**) Western blot analysis of EGFR, ERK, and AKT total or phosphorylated and cleaved PARP. EGF ligand was used at 10 ng/mL for 10 min. CTR: control; AFA: afatinib, ALI: allitinib at 0.2 μM.

**Figure 2 ijms-23-07774-f002:**
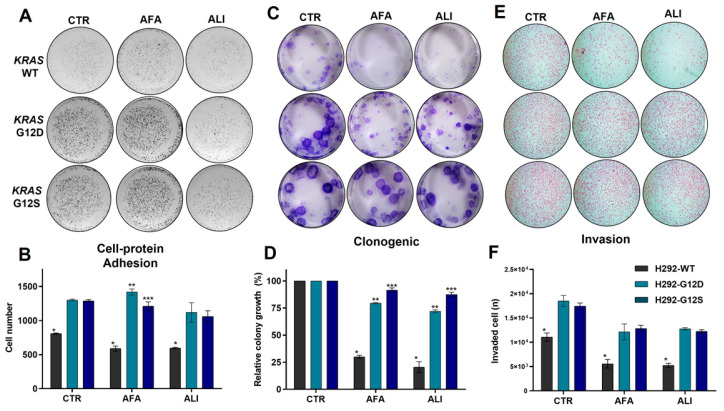
H292 *KRAS* wild-type and mutant cell lines were exposed to both EGFR inhibitors at 0.2 μM. Cell–protein adhesion assay (**A**) and bar graphs indicate the number of attaching cells (**B**); clonogenic assay (**C**) and bar graphs indicate the relative colony number (**D**); invasion assay (**E**) and bar graphs represent the relative number of invading cells (**F**). Cells at 10× magnification. CTR: control; AFA: afatinib, ALI: allitinib. * *p*-values < 0.01, ** *p*-Value < 0.001, *** *p*-value < 0.0001.

**Figure 3 ijms-23-07774-f003:**
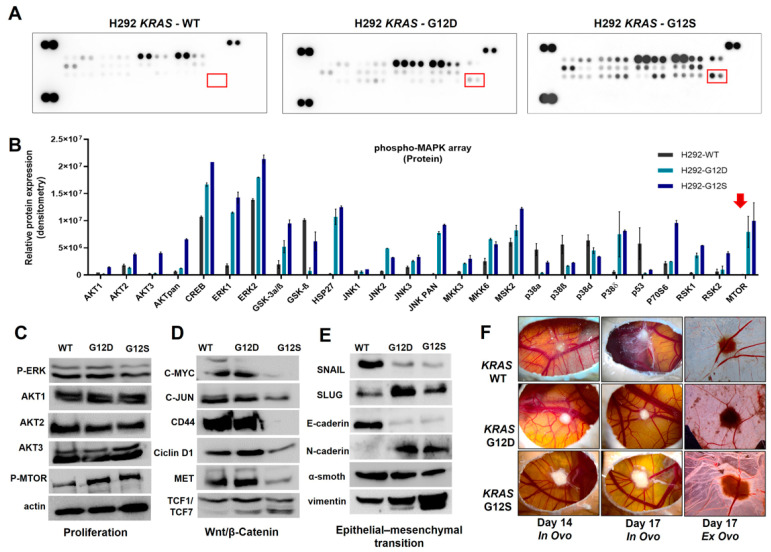
Molecular alterations in *KRAS* mutated cell lines. Representative pictures of phospho-RTK arrays for H292 *KRAS* wild-type, *KRAS* (G12D), and *KRAS* (G12S) cell lines (**A**). Each RTK is duplicate in the arrays (two spots side by side) and four pairs of phosphotyrosine positive controls in the corners of each array. Densitometric analyses are represented by bar graphs and red arrow show mTOR in KRAS mutated cells. Red squares show the mTOR phosphorylated levels (**B**). Analysis of proliferation markers (**C**), Wnt/β-catenin (**D**), and epithelial–mesenchymal transition (**E**) markers by Western blot. In vivo effect of KRAS mutations on H292 cell line growth (**F**). Representative pictures (×16 magnification) of CAM assay in ovo and ex ovo at 14 and 17 days. Data presented as a mean of 18 eggs per group.

**Figure 4 ijms-23-07774-f004:**
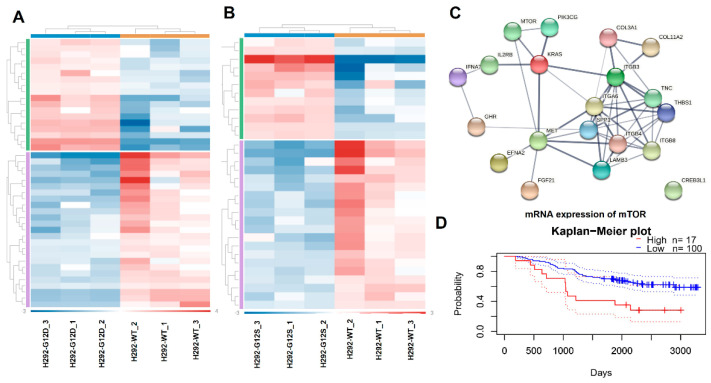
Gene expression analysis of a pan-cancer panel by NanoString™ and functional pathways analysis. Heatmap of genes altered in H292-KRAS mutated cells H292 *KRAS*-G12D (**A**) and H292 *KRAS*-G12S (**B**) compared to H292-*KRAS* wild-type. In red represents the overexpressed genes, and in blue, the down expressed genes. Genetic interaction network associated with *KRAS* mutations on String platform (**C**). Each circle represents a gene (node) in this figure, and each connection represents a direct or indirect connection (edge). Kaplan–Meier plot survival of mTOR expression in a cohort of NSCLC patients (*n* = 117) by PrognoScan analyses (**D**).

**Figure 5 ijms-23-07774-f005:**
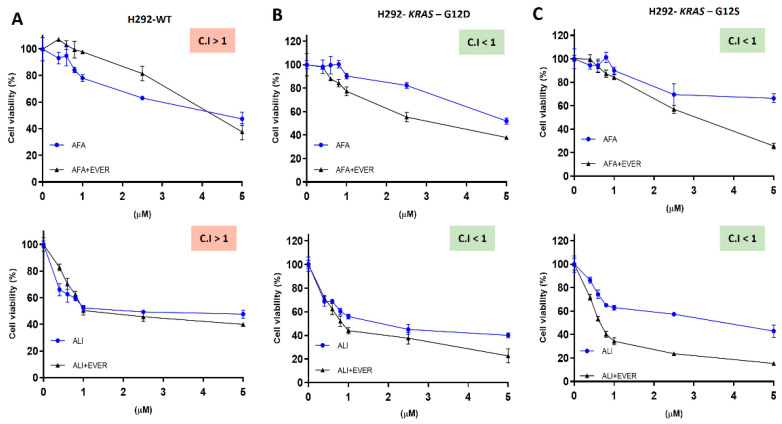
Cell viability assay of wild-type-WT (**A**) *KRAS* mutated cell G12D (**B**) and *KRAS* mutated cell G12S (**C**) exposed to EGFR inhibitors in combination with everolimus (1 μM) for 72 h. Squares represent combination index (CI) values. Synergy (CI < 1.0); antagonism (CI > 1.0); and additivity (CI = 1.0). AFA: afatinib, ALI: allitinib, EVER: everolimus. Data presented as the mean of three independent experiments.

**Table 1 ijms-23-07774-t001:** Mutation analysis and anti-EGFR drug response of NSCLC cell lines.

Cell Line	IC_50_ ± (SD) µM				Mutation Status				
	Allitinib	Afatinib	Lapatinib	Erlotinib	*KRAS*	*NRAS*	*PIK3CA*	*EGFR*	*ERBB2*
NCI-H1975	0.21 ± 0.09	0.34 ± 0.03	20.9 ± 5.91	18.3 ± 3.94	WT	WT	p.G118D	L858R + T790M	WT
HCC-827	0.31 ± 0.07	0.51 ± 0.13	9.68 ± 0.27	11.8 ± 1.12	WT	WT	WT	del19	WT
PC9	0.29 ± 0.03	0.72 ± 0.05	8.49 ± 0.51	10.2 ± 0.39	WT	WT	WT	del19	WT
SK-MES-1	0.96 ± 0.10	1.51 ± 0.52	30.3 ± 2.52	36.07 ± 1.95	WT	WT	WT	WT	WT
SK-LU-1	0.87 ± 0.07	2.16 ± 0.63	31.24 ± 4.05	48.02 ± 5.15	p.G12D	WT	WT	WT	WT
A549	3.92 ± 1.03	5.18 ± 1.02	25.23 ± 2.98	36.42 ± 4.03	p.G12S	WT	WT	WT	WT
NCI-H292	1.32 ± 0.94	3.96 ± 0.98	27.8 ± 3.15	>50	WT	WT	WT	WT	WT
COR-L23	3.27 ± 1.08	6.34 ± 0.72	>50	>50	p.G12V	WT	WT	WT	WT
COR-L105	0.98 ± 0.76	0.92 ± 0.12	1.39 ± 0.79	2.22 ± 0.92	WT	WT	WT	WT	WT
LUDLU-1	7.35 ± 1.26	2.47 ± 1.05	>50	10.4 ± 2.19	WT	WT	WT	WT	WT
NCI-H322	1.54 ± 0.89	0.35 ± 0.03	12.81 ± 3.05	6.19 ± 1.17	WT	WT	WT	WT	WT
NCI-H358	1.24 ± 0.65	0.81 ± 0.04	12.25 ± 2.98	8.32 ± 2.01	p.G12C	WT	WT	WT	WT
NCI-H727	1.65 ± 0.65	0.42 ± 0.06	6.34 ± 1.12	9.64 ± 2.45	p.G12V	WT	WT	WT	WT
NCI-H2228	1.50 ± 0.01	3.98 ± 0.02	>50	30.0 ± 0.71	WT	WT	WT	WT	WT
Calu-3	4.52 ± 0.31	7.12 ± 0.80	>50	>50	WT	WT	WT	WT	WT

## Data Availability

Not applicable.

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
