# Peer review of "Efficacy of Combined Use of Everolimus and Second-Generation Pan-EGRF Inhibitors in KRAS Mutant Non-Small Cell Lung Cancer Cell Lines"

_ijms, 2022, doi:10.3390/ijms23147774_

Round 1

Reviewer 1 Report

Thank you for the chance you gave me to read this interesting study entitled “Efficacy of Combined Use of Everolimus and Second-generation Pan-EGRF Inhibitors in KRAS Mutant Non-small Cell Lung Cancer Cell Lines” by Silva-Oliveira et al. In this original research paper, the authors evaluated the efficacy of pan-EGFR inhibitors in a panel of 15 lung cancer cell lines in association with KRAS mutations phenotype. In addition, the combination of the same EGFR inhibitors with everolimus was also studied. This is a very interesting study presenting pre-clinical data regarding the treatment of NSCLC harboring KRAS mutations. Although, this topic has great importance, however, there are many significant issues with this manuscript. I think that this study in the current form doesn’t satisfy the appropriate criteria for publication.

Some of them are:

According to the score (36%) obtained by the plagiarism detection service “Turnitin”, the manuscript needs to be modified in some parts in order this score to be reduced.

The study needs extensive editing of English language since in some sentences the meaning is obscure and the text doesn’t flow

“Results” section could be improved with better organization of the available data.

Minor:

All abbreviations should be expanded at their first mention.

Line 34: Please, clarify the term “sensibility”.

Line 41: “Postponed” should be replaced.

Line 49-50: The phrase “70% are amenable to complete surgical resection” refers only to patients with adenocarcinoma. Is this what the authors want to say?

Lines 53-54: Please, rephrase.

Line 56: Please, replace del19 with exon del 19 since the first doesn’t make sense.

Initial letters of drug names should be written with lowercases e.g. Allitinib line 60.

The authors should provide information on the role of the third-generation EGFR TKIs in the first line of treatment in the introduction section.

Lines 80-82: Please, provide the most updated data regarding adagrasib (see ASCO2022).

Author Response

Thank you for the positive response to our manuscript “Efficacy of Combined Use of Everolimus and Second-generation Pan-EGRF Inhibitors in KRAS Mutant Non-small Cell Lung Cancer Cell Lines- ijms-1778488” by Renato José da Silva-Oliveira and co-authors. The manuscript has been reviewed in accordance with the reviewers and editorial board's suggestions. All the corrections in the manuscript were marked with word track changes in the last manuscript vision uploaded to the website as recommended. Below please find a detailed 'point-by-point' response to the reviewers' comments.

Reviewer 1

Major comments

  • According to the score (36%) obtained by the plagiarism detection service “Turnitin”, the manuscript needs to be modified in some parts in order this score to be reduced.

Response: We appreciate the reviewer's analysis. We conducted an extensive analysis of the main text and changed it to reduce the plagiarism percentage.

  • The study needs extensive editing of the English language since in some sentences the meaning is obscure, and the text doesn't flow.

Response: We carefully read the manuscript to correct spelling errors.

  • “Results” section could be improved with better organization of the available data.

Response:  We performed an extensive number of in vitro and in vivo assays to show the role of mTOR in intrinsic resistance to EGFR inhibitors using KRAS mutant NSCLC cell lines. Therefore, we separated the findings into four topics to elucidate the main experimental findings, starting from the efficacy of different EGFR inhibitors, passing through phenotype and molecular changes caused by KRAS mutations, and ending with the mTOR protagonist in KRAS resistance mechanic.

Minor comments

  • All abbreviations should be expanded at their first mention.

Response: We reviewed the entire manuscript and completed the suggested modifications in abbreviations

  • Line 34: Please, clarify the term “sensibility”.

Response: We rewrote the sentence. "Significantly, everolimus restored sensibility and improved cytotoxicity of EGFR inhibitors in KRAS mutant NSCLC cell lines context.”

  • Line 41: “Postponed” should be replaced.

Response: We rewrote the sentence. "The diagnosis is usually late and is made in the advanced stages, leading to inefficient curative treatment."

  • Line 49-50: The phrase “70% are amenable to complete surgical resection” refers only to patients with adenocarcinoma. Is this what the authors want to say?

Response: Correct, we refer in this line that 70% of adenocarcinomas are amenable to complete surgical resection

  • Lines 53-54: Please, rephrase.

Response: We rewrote the sentence: "Complete surgical resection is amenable in 70% of adenocarcinoma [6]; 25 to 30% in squamous cell carcinomas, and just over 10-15% in large cell carcinomas or undifferentiated [7].

  • Line 56: Please, replace del19 with exon del 19 since the first doesn’t make sense.

Response: We rewrote the sentence.

  • Initial letters of drug names should be written with lower cases, e.g., Allitinib line 60.

Response: We reviewed the entire manuscript and changed all incorrect names.

  • The authors should provide information on the role of the third-generation EGFR TKIs in the first line of treatment in the introduction section.

Response: We rewrote the section and included the sentence. "[14]. In addition, results from the Phase III AURA3 trial demonstrated the superiority of osimertinib over standard platinum-based treatment of NSCLC patients with EGFR T790M mutation relapsed or refractory to first-line EGFR TKI therapy, thus definitively establishing this third-generation TKI as current standard treatment [15]. Other third-generation of EGFR TKIs, such as EGF816, olmutinib, PF-06747775, YH5448, avitinib, and rociletinib, continue to be investigated in clinical trials phases to prove their alone or combinates therapeutic efficiency [16].

  • Lines 80-82: Please, provide the most updated data regarding adagrasib (see ASCO2022).

Response: We rewrote the introduction section at 86 – 89 lines. “Moreover, adagrasib (MRTX849), also targeting KRAS G12C-mutated tumors, is currently being evaluated in phase 1/2 with results resembling the efficacy of sotorasib [20]. A recent update by the KRYSTAL-1 trial showed tolerable levels (600 mg twice daily) and demonstrated durable (16.4 months) clinical activity in NSCLC patients KRAS G12C mutant [21], and added a phase 3 trial (NCT03785249) evaluating adagrasib as monotherapy versus docetaxel in KRAS mutant NSCLC patients is ongoing.

            Sincerely,

Rui Manuel Reis, PhD

Molecular Oncology Research Center, Barretos Cancer Hospital

Rua Antenor Duarte Villela, 1331 - CEP 14784 400, Barretos, S. Paulo, Brazil

Phone/Fax:+551733216600 - Extension: 7090

Reviewer 2 Report

Clinically, EGFR and KRAS mutations are often present simultaneously in lung cancer, and as the authors say, drug resistance is very problematic.

The authors performed protein array and other assays on lung cancer cells. The results showed that inhibition of mTOR with EGFR TKI suppressed the proliferation of lung cancer cells.

I consider this to be a necessary and sufficient experiment.

Author Response

Thank you for the positive response to our manuscript “Efficacy of Combined Use of Everolimus and Second-generation Pan-EGRF Inhibitors in KRAS Mutant Non-small Cell Lung Cancer Cell Lines- ijms-1778488” by Renato José da Silva-Oliveira and co-authors. The manuscript has been reviewed in accordance with the reviewers and editorial board's suggestions. All the corrections in the manuscript were marked with word track changes in the last manuscript vision uploaded to the website as recommended. Below please find a detailed 'point-by-point' response to the reviewers' comments.

Reviewer 2

Comments

Clinically, EGFR and KRAS mutations are often present simultaneously in lung cancer, and as the authors say, drug resistance is very problematic.

The authors performed protein array and other assays on lung cancer cells. The results showed that inhibition of mTOR with EGFR TKI suppressed the proliferation of lung cancer cells.

I consider this to be a necessary and sufficient experiment.

Response: We appreciate for the very positive assessment of our study.

            Sincerely,

Rui Manuel Reis, PhD

Molecular Oncology Research Center, Barretos Cancer Hospital

Rua Antenor Duarte Villela, 1331 - CEP 14784 400, Barretos, S. Paulo, Brazil

Phone/Fax:+551733216600 - Extension: 7090